# Sustainable Land-Use Pathway Ranking and Selection

**Garth John Holloway**

Department of Applied Economics and Marketing, School of Agriculture, Policy and Development, University of Reading, Whitenights, P.O. Box 217, Reading Berkshire RG6 6AH, UK; garth.holloway@reading.ac.uk

**Abstract:** The desire for refining status quo cost–benefit protocols to fully encompass econometric model uncertainty motivates the search for improved technology. Availability of unique Ethiopian highlands milk-market livestock data provides an ideal laboratory for investigation of alternative land-use pathway designs. In these contexts, we present novel methodology for ranking and selecting sustainable 'land-use pathways', arguing that the methodology is central to sustainable-land-use-policy prescriptions, providing essential innovation to assessments hitherto devoid of probabilistic foundation. Demonstrating routine implementation of Markov-Chain, Monte-Carlo procedure, ranking-and-selection enactment is widely disseminable and potentially valuable to land-use policy prescription. Application to a sample of Ethiopian-highlands, land-dependent households highlights empirical gains compared to conventional methodology. Applications and extensions that profit future land-use sustainability within the Ethiopian highlands and, also, more generally, are discussed.

**Keywords:** sustainable-land-use pathway ranking-and-selection; Markov-chain Monte-Carlo methodology; Ethiopian-highlands-land-dependent case study

## 1. Introduction

As noted, occasionally, within recent contributions [1–4], human–livestock interactions are one of the major contributing forces effecting global land-use change. Moreover, as noted [4] (p. 1), "The livestock sector is a key element of land-related human interference ... [and] ... Accordingly, restraining land requirements is, increasingly, regarded as a decisive measure to alleviate detrimental impacts of livestock production on the environment ... ".

The usual tool applied for informed evaluations of the impacts of land-use change is the conventional method of cost–benefit analysis (which originates from Jules Dupuit, circa 1848, being, later formalized by Alfred Marshall). Given this long history, there have been many modifications of the basic ideas. Increasingly, elements of risk and uncertainty have been recognized as essential fulcrums with which cost–benefit foundations are more suitably applied. Integrative risk and uncertainty components of empirical cost–benefit assessments are now considered one mainstay of analysis [5]. Increasingly, however, authors have grown to recognize that considerable model uncertainty prevails across the plethora of regression settings from within which cost–benefit analyses are typically enacted [6–10]. See, for one very interesting, quite recent, tour across essential elements, combining machine learning, the contribution within [11]. Thus, consideration of alternative livestock profiles combines uncertainties not only inherent from within the agro-ecological, human–animal interface, but also evolving from subjective choices imparted the investigation by the human–investigation interface, encompassing debate.

A situation such as the one presented could be evaluated, formally, by the application of so-called multi-criteria-decision-making protocols such as the so-called analytic-hierarchy processes or, alternatively, and arguably, more generally, by applying the analytic network processes [12].

These procedures, while robust to some settings, have the disadvantage that one of the inputs used to delimit the decision-making process is the subjectively assigned empirical setting. By way of example, one such situation ariese and is discussed by [5]. In these contexts, among others, the principal source of uncertainty derives from the choice of model formulations used to produce the empirical inputs applied by the decision-maker. In deriving estimates of key parameters and, thus, having a procedure for adequately representing this uncertainty present and embedding it within the decision-making process is one attraction of the alternative methodology. In this context, Markov Chain Monte Carlo procedures prove indispensable for navigating the rich and varied model space confronting the investigation. These inputs call forth some new demands, which can, of course, present new challenges. Fortunately, however, a rich set of implementation protocols (see, for example, [13–15]) avail themselves for implementation. For one quite recent review of the significance of the Hastings algorithm, in this regard, see [16].

In short, a substantial research gap exists, but has remained mostly latent in previous work. The research gap becomes increasingly significant once model uncertainty is embraced, fully. The research gap existence raises scope for nuanced empirical investigation. The research gap is further highlighted by the fact that we know very little about procedures for combining these types of uncertainties and for informing empirical cost–benefit analysis. Progress in this direction could lead to abundant windfalls. The windfalls that are likely to arise evolve from the discernment of appropriate policy prescriptions hitherto ignored or left nascent by conventional methodologies and the the likely gains accruing implementation of improved cost–benefit procedures that take fuller account of all prevailing and relevant empirical uncertainties. Locating one such beneficial procedure is the task presented before us and is the task with which we engage in the remainder of this contribution.

This paper presents a procedure for ranking and selecting alternatives in the general context of policy evaluation and the specific context of land-use prescription within which sustainable livestock systems may co-exist [17]. The new methodology enhances traditional cost–benefit analysis extending investigation in order to rank and select policy options with due regard for stochastic context. We showcase the procedure in a development-economic setting where the policy options impact land-use productivity, stem institutional innovation, and mitigate potential barriers impeding household welfare. The methodology has potentially broad application whenever policy analysis involves econometric investigation. We argue that this situation is common. Because the results of econometric investigation rely upon unobservable quantities, a problem of comparing probability distributions arises. Attendant complications emanate from investigator-ascribed, possibly heterogeneous, subjective evaluations of the costs and benefits of alternative policies. In short, complexity confronts stochastic policy evaluation, in general, and land-use-policy prescription, in particular, and it is useful to avail methodology for informing decision-making within this circumstance.

We demonstrate how measurement problems are resolved, straightforwardly, as an outcrop of conventional methodology, exploiting extant Markov-Chain Monte-Carlo (MCMC) methods. 'MCMC' is a broad class of iterative procedures for estimating non-conjugate models in the context, primarily, of Bayesian investigations. MCMC is now in fairly widespread dissemination amongst practitioners. Yet, despite perceived advantages, and, to the best of the author's knowledge, land-use policy neglects stochastic ranking-and-selection methodology. Scope exists for additional enquiry.

Impact assessment investigations, in general, and the empirical setting of this paper, specifically, encounter substantial additional uncertainties when prescribing ranking and selection pathways. Uncertainties, in at least three dimensions, typically, confront investigation. Uncertainty arises about the appropriate sets of covariates to include; uncertainty arises about the appropriate empirical specification to enact; and uncertainty arises about the relevance of 'simultaneities' across 'space', across 'time', or across 'the agents' (in this case, households) generating the observed data. One unfortunate encounter arising from ad hoc judgment is that 'nature' may assign negligible 'support' to arbitrary formulations, motivating search. To the extent that policy prescriptions are often affected by arbitrary choice, there is concern for the overall robustness of the econometric procedure, the empirical

conclusions drawn from the exercise, and the precision of predictions so derived. Frequently (although less often during Bayesian investigations), little attention is devoted to the problem of 'search'; and the absence of such devotion generates additional scope for nuanced econometric enquiry.

We highlight the importance of ignoring 'search' during the process of investigation, demonstrating the ways in which policy prescriptions are affected by ignoring available 'evidence'. We demonstrate that some important conclusions about land-use pathway choice remain largely unaffected by this neglect, whereas others are more sensitive and ultimately over-arch the ranking-and-selection exercise. In short, we consider, comprehensively, the main uncertainties confronting land-use pathway policy prescription; incorporate allowance for these uncertainties; and evaluate each one, formally, by model-averaging the available 'evidence' supporting them.

We showcase the fundamental role of the search procedure after discussing the basic motivation underlying the 'new' methodology, which we present in Section 2. In Section 3, we introduce the empirical investigation. Section 4 details the empirical search procedure and presents its results. Section 5 introduces the central policy question motivating the empirical search and presents empirical results. Section 6 presents the empirical ranking and selection procedure and its results. Section 7 highlights the main contributions, discusses the main limitations, identifies further applications that might be possible, and details extensions of the present effort that are particularly relevant to land-use policy design. Section 8 concludes.

## 2. Motivation

The motivations underlying the statistical investigation evolve from extant frequentist approaches [18–20]. Bayesian influence evolves, first, from [21] and, subsequently, from [22]. The primary motivation is 'ranking' a set of unknown means and 'selecting' one of the means as 'best'. The problem is essentially one of ascertaining, formally, 'the evidence' in favor of the outcome that one of the means dominates the choice set being ranked 'highest' or 'lowest' among the set of candidates. Informal approaches to this problem often amount to selecting 'the largest' or 'the smallest' quantity. This approach is not well supported. When variances are disparate, or, more significantly, when the posterior densities of the relevant continuous random variables differ, or the probability mass points of the relevant discrete random variables, differ, this approach produces erroneous conclusions.

As an example, highlighting the difficulties belying comparisons of means from different populations, [22] presents (Table 1, p. 365) relating batting averages in baseball and the results of such choice. They demonstrate formally just how conventional methodology fails to account adequately for the probability support underlying actions and how it leads to 'false' conclusions (conclusions that differ dramatically from the ones espoused by formal probability calculations). The case considered by [22]'s 1988 baseball example motivates three aspects of the ranking-and-selection procedure applied subsequently. First, differences in distributions significantly affect the rankings. Second, failing to account for differences leads to false conclusions. Third, Bayesian procedures yield informative content, which classical ('frequentist') methodology ignores.

The practical implications of the procedure can best be considered with the introduction of a case-study and a brief review of some background work surrounding the empirical investigation.

## 3. Background to the Empirical Investigation

Broadly speaking, our ranking-and-selection empirical study embraces alternative pathways for achieving one specific economic goal, deemed beneficial for sustainable productivity enhancement, private and public economic benefit, and, ultimately, the alleviation of poverty among subsistence households.

'Time' is one of the most significant resources available to the household [23]. This idea—the idea that 'time available to the household' is a key resource—has spawned substantial literature surrounding household decision-making [24,25], household production in subsistence-economic settings, intra-household and inter-household interactions [26,27], and related, miscellaneous

application of these ideas [28]. Notwithstanding this perspective, it is equally likely that among the most important, resources available to the household, is the total amount of land that the household has available for allocation to its 'enterprise'. It is not coincidental, therefore, that the time-resources available and the land-resources available to the household feature significantly in the household-production-model framework [26].

Some encounters in developmental economic settings in which land-use property rights are deemed important, and are thought, often, to over-arch the realms of other economic decision-making, arise in rural Ethiopia. In the rural Ethiopian land-use setting [29–37], resources are essentially state assigned and implicate, in important and sometimes unsavory ways, features of household decision-making. The authors of [29] provides a general background to the evolution of land-use struggles in rural Ethiopia; [30] details recent directions and change; [31,32] present examples of land-use policy analysis using recently collected Ethiopian data; [33–35] extends and formalizes these ideas; and [36,37] collates and assimilates central ideas in land-use decision-making and in land-use pathway choice.

Generally speaking, land-use policy debates within the Ethiopian, subsistence-household context surround the question of how best to use the land in order to achieve stated policy objectives. Broadly speaking, these objectives relate productivity (see, for one especially relevant example intimately related to present intent, [38]), and sustainable development and poverty alleviation (see, for general, but especially important, examples, [39]); but are, sometimes, modified in the face of contemporaneous political objectives.

One issue at the forefront of the minds of Ethiopian policy-makers is how best to effect entry into markets when the markets themselves are 'nascent', or, better, 'latent', and are constrained by 'thinness' or 'lack of density of participation' and contain attendant 'instability' [2,38]. The term 'market precipitation' is sometimes used in order to convey the notion of 'solidifying' atomism and 'connecting' agents ('Precipitation' is used in scientific settings to denote the creation of a solid from a solution, whereby, the solid formed when a reaction occurs is sometimes referred to as 'a precipitate'.). Market incumbency, or, rather, non-incumbency, is a recurrent theme transecting developmental objectives, generally, but, especially throughout sub-Saharan Africa (see, for example, [2]; and more recently, [38]). This focus makes our Ethiopian case study 'somewhat-ideal' and a somewhat credible 'social-science laboratory'.

In the presentation to follow, we enact the [22] ranking-and-comparison procedure, extending their approach in several important ways. The exercise is instigated expressly for the purpose of comparing alternative means for promoting non-participant entry into nascent milk-markets. We employ household production and marketing data collected at sites close to Addis Ababa, at three visits during the 1997 milk-production year. Using the sales data, we formalize MCMC methodology for inferring the 'distances' that households reside from the market; measure these distances in terms of essential covariates; and, subsequently, use costs information to rank and select 'best policy' for promoting 'precipitation'.

We enact cost–benefit policy evaluations in two steps. First, we measure the extent of departure by the household from the marketplace by converting the latent distance measures in the dependent sales quantity to measures of resource deficits across key covariates. Given these resource deficits, we obtain appropriate cost information and use the [22] ranking and selection procedure to determine 'best' policy. Although the estimation procedure is routine, we find no previous application of the idea within a developmental setting. Hence, the study may be the first of its kind, but is almost surely (and, genuinely, to the best of the author's knowledge) 'seminal', with respect to developmental initiatives. At a minimum, the study provides one novel intervention from which further refinement evolves.

Finally, as in any empirical investigation, the econometrician confronts substantial uncertainties concerning which of, a countably finite, but, potentially, limitless number of formulations to enact; the specific covariates that might affect production, consumption, and sales decisions undertaken by the households; and additional formulaic nuances concerning the data generating process. Accounting

for the considerable model uncertainty confronting application of the Ethiopian household data is undertaken with reference to 'the evidence', which is the fundamental quantity appearing on the left side of [40]'s famous (1937) identity. Formally, given data, $\mathbf{y}$; unknown quantities, $\theta$; and probability density functions $f(.)$; the quantity $f(\mathbf{y}) \equiv \int f(\mathbf{y}|\theta) \times f(\theta) \, d\theta$, promotes the left-side quantity as the integration of the right side quantities (within which 'common science' prevails) and is otherwise known as the 'marginal likelihood' or 'the evidence' or 'the marginal-density' for the data [41,42]. The identity makes clear our investigation, where, for the bulk of initial analysis, we linger in the left side, before moving into the right side subsequent to formaing conclusions concerning model formulation. In terms of the [40] identity, our model-search procedure is relatively straightforward. We compute quantities $f(\mathbf{y})$ for each 'model' under consideration and use these quantities to make probabilistic statements about the possibility that a particular 'model' generated the data [41,42].

We define 'model', within this setting, to denote choices in three dimensions, namely, a particular choice of covariates, a particular choice of censoring thresholds characterizing milk-market immersion, and a particular form of geo-spatial quantities characterizing socio-economic interactions. We then apply these probabilities to the ranking-and-selection procedure, extending [22] in order to construct a (model-averaged) ranking and selection protocol as the basis for our final conclusions. Turning attention once again to the baseball batting-averages example, Ref. [22] used a single modelling assumption to highlight conclusions. In [22], the investigator confronts substantial uncertainty in addition to the ranking and selection outcome. In [22], additional potential models that could have generated the data are ignored; presently, they are not. Finally, in order to highlight the potential folly of ignoring alternative models, we enact comparisons with common procedures, and the model of choice in the seminal Tobit, censored regression arising at the advent of the Gibbs-sampling revolution, namely [43].

A preliminary survey was undertaken in 1995; the Ilu-Kura and Mirti 'peasant associations' were selected for data collection; and, in 1997, a survey of household demography, production, and sales was undertaken. In total, 68 household records are available for evaluation, with 35 records sampled from the Mirti Peasant Association and the remaining 33 derived from the Ilu-Kura Peasant Association [44].

## 4. Search

The 68-household records constitute a fairly robust sample of data available for analysis of a variety of features of fundamental significance within the household-production setting. The over-arching interest in these data stems not from their clandestine geo-spatial, socio-economic contexts; but, in the extent to which their sequestered representation exemplifies features of the sample setting (animal and land-use productivity specifically; East-African dairying, generally; and market immersion, thematically) that have relevance in a wide and broader set of circumstances. Presently, of course, interest lies, ostensibly, in other matters.

We are primarily concerned with the data's usefulness as a motivation for land-use policy prescription, land-use pathway ranking and selection, and as a motivational input for characterizing extant methodology, namely, Ref. [22] and its extension. In this setting, and, because previous work with components of the data, multilaterally ignores model uncertainty; it is pertinent, relevant, and informative to identify, briefly, previous investigations that have used components of the survey and what they have achieved.

Early work with the data [45] investigates strategies for promoting 'agro-industrialization' of East-African dairy production; considers the potency of 'local-' and 'global-based' policy interventions; applies conventional Tobit methodology; and treats the full (68-households × 3-visits × 7-days-sales-recall =) 1428 sample observations as a single 'cross-section', ignoring hierarchical structure. In [46], background theory and conceptual frameworks are presented for implicit-function analysis and related motivations for multi-variate extensions of the single-equation methodology in [45]. In [47], we consider the production data and investigate finite mixtures technologies. In [48–50], we investigate household willingness-to-pay for extension services applying conventional probit

methodology; model market entry as a binary choice; and implement the full 1428-observation panel as a single, unified cross-section. In [51–53], we study cross-bred cow adoption using single- and multi-variate investigative structures; use a latent Gaussian simulator for identifying discrete amounts of the cross-breed cow quantities; model joint decision-making across the 204-observation sample subset for which adoption, production, and marketing quantities exist; and apply conventional multi-variate-Normal linear modelling frameworks, conventional censoring, and non-conventional count-adoption technology. In [54], we consider random censoring of the Tobit regression; adapt ordered-probit methodology ([55]) to the Tobit; and enact investigation applying the 1428-observation panel as a single, unified cross-section. Lamenting the paucity of MCMC investigations in development-economic settings at that time, [56,57] summarize key findings from the survey; present pedagogic introductions to Gibbs sampling in normal-linear, censored-regression, and binary-choice frameworks; and suggest directions for future applications of Gibbs sampling, in particular, and MCMC technology, in general, in development-economic settings. In [51], we perform meta-analysis on market immersion, participation, and adoption studies in the East-African setting and beyond; and review previous work with the survey data. In [48], we present an initial conceptual framework for impact assessment; apply computations to a conventional Tobit setting; and set a stage for further extension of the [22] methodology. In [54], we compare and contrast three, alternative censoring structures for improving inference in the Tobit censored regression when there is belief that the censoring point is not zero; improve upon the initial approach presented in [45]; and apply single-equation procedures to the 1428-observation sample ignoring additional household heterogeneities. In [51,52] we consider two discrete decisions, namely whether to participate and by how much; model this additional barrier to entry using extant probit MCMC methodology ([55]); and apply the procedure to a 204-observation subset of the survey data. In [56] and in [57], we extend a rich literary heritage on transactions costs, formal household-production frameworks, the Kuhn–Tucker conditions characterizing 'corner solutions' and develop a joint-decision-making, random-censored household-production model of the sales-versus-production-versus-adoption decisions characterizing the household frameworks. Finally, [58] employs a multi-dimensional, hierarchical framework combining the diverse panel for the 1428-observation sales quantities; the 204-observation production quantities; and the 68-observation socio-demographic sample in order to accurately assess marginal-productivity differentials across two genetic resource stocks (indigenous- and exotic-breed animals) in order to measure accurately costs of catastrophic livestock loss.

Each of these studies makes incremental contributions; is linked by due concerns for poverty alleviation; inter-connected by applications of continual improvements in Gibbs sampling, specifically, and MCMC, in general; and encompasses other features of relevance to the socio-economic setting. However, one additional linkage is negative. This linkage is that each of the aforementioned studies fails to take formal account of model uncertainties. In particular, the covariates considered; the censoring frameworks under evaluation; and a host of additional, neglected aspects of possible significance to 'representation', 'inference', and 'prediction' are either ignored or are chosen arbitrarily, subjectively and in an ad hoc manner. To make this criticism more cogent, it is apparent, and, nowadays, perhaps unforgivable, that no single empirical interventions within this unified set of intellectual contributions enacts formal, marginal likelihood comparisons.

Unsurprisingly, previous work with the data predisposes the investigation to some fairly strong *a priori* sentiments directing model construction. However, previous work fails to account adequately for the form of uncertainty one should like to evaluate, fully, and incorporate, comprehensively, for the purpose of drawing robust policy conclusions. Inevitably, the alleged inadequacies of previous work—alleged inadequacies that we seek to formalize subsequently—raise considerable scope for additional scrutiny and extended application.

Previous Tobit interventions focus attentions on seven covariates deemed 'important', moreover, 'fundamental' to previous enquiry. These covariates are, specifically, the household's proximity to the milk-cooperative (so-called 'timetothemilkgroup'); the observed number of years of formal education



of the household head ('education'); the observed number of cross-breed cattle available for milking ('crossbreed'); the observed number of indigenous-breed cattle available for milking ('localbreed'); the observed number of extension visits ('extension'); and two site-specific dummy variables ('ilukura' and 'mirti'). Among these chosen covariates, interest centers on the relative potency of extension visitation and indigenous- and exotic-milking-herd formation as catalysts for enhancing intellectual and physical capital of the household, elevating production capacity, and effecting entry into milk markets.

## 4.1. Covariate Search

These seven chosen covariates, we emphasize, are, of course, chosen with substantial *a priori* information about the sample setting, the production practices of the households, and the policy prescriptions confronting policymakers. Notwithstanding this remark, this seven-covariate choice is, of course, a purely subjective choice. Further, this choice, inevitably, raises scope for re-examination of the appropriate covariates guiding production, sales, and marketing decisions across the Ethiopian sample. In order to illustrate the significance of this choice, we enact model comparisons across an expanded subset of some 20 covariates. The full list of 20 covariates includes the preceding seven plus the additional 13 quantities: The gender of the household decision-maker ('gender'); a proxy for the transportation potential within the household (constructed by normalizing the number of children of transportable age within the household by the number of household members, 'transportproxy'); the distance (measured in kilometers) to the nearest road ('roaddistance'); the distance (measured in kilometers) to the nearest market ('marketdistance'); a binary indicator representing credit use ('credituse'); the number of years of experience in farming of the household head ('experience'); the time spent waiting at the milk group ('timewaitingatthemilkgroup'); the total amount of crop land available to the household ('cropland'); the total amount of pasture land available to the household ('pastureland'); a binary indicator depicting ownership of farm equipment ('ownfarmequipment'); a binary indicator depicting ownership of transport equipment ('owntransportequipment'); and a constant ('constant').

A set of covariate-selection protocols is available from adapting a powerful algorithm developed previously by [59]. The covariate-selection exercise (for which all of the computer code is available as an online Appendix A to this document) makes clear a fundamental fact, indicting previous applications with the data. This fact is that, given 'K' possible covariates, once the full $2^K$ model space is accessed, it appears that there is little support for the arbitrarily chosen seven-covariate design matrix used previously by the author and collaborators. Inevitably this low support (low from a marginal-likelihood, probability perspective) draws into question the implications of inferences derived from previous investigation. The present investigation cannot ignore the randomly selected covariate approach, which vastly dominates the alternative. We present evidence supporting this conclusion, subsequently.

## 4.2. Threshold Search

In addition to covariate uncertainty, considerable uncertainty arises concerning the form of 'censoring' that is appropriate for each respective household. Within non-participating households the actual level of marketable surplus that the household engenders is 'latent'. Consequently, the observed data are then left censored. Traditionally, the marketable-surplus censor point is zero. However, theory suggests otherwise. It is more realistic to assume that there exists some positive amount of marketable surplus beyond which trade becomes feasible. Thus, censoring, in general, and the censoring point, in particular, are important foci. Hence, we estimate the censoring threshold as another parameter within the censored-regression framework.

In order to assess the implications of the random-versus-zero-censor point, we resurrect the covariate-selection exercise implemented above and compute the log-marginal-likelihoods for each of the alternative models, when the censoring point is permitted to vary randomly between the lower bound of zero and the upper-bound of one (the minimum sales quantity computed across the 68-household sample is observed to be exactly one liter of milk per household per day). For each of

the models considered in the initial exercise we observe that the random-censored Tobit regression out-performs its zero-censored counterpart, but not substantially. The dominant model is, once-again, the random-selection covariate setting using the algorithm formalized in [59]. This preferred model has dominant probability support in comparison with the other model, for which the probability support is non-negligible. We conclude, therefore, that it is prudent, moreover motivated by the probability calculations, that the investigation proceeds with due regard for, both, the random- and the zero-censored formulations. We present evidence supporting this conclusion, subsequently.

### 4.3. Neighborhood Search

Finally, within the context of searching for appropriate specification, neglect of so-called neighborhood effects [60] is questionable. In the background, seminal theory, on the allocation of 'time' within the household, we also learn of the importance of social interactions [23]. Additionally, the author has informal casual empirical experience (obtained in visits to the survey sites, circa 1998) to suggest that there may be strong influences across connected 'neighbors' at both survey sites. The reader is reminded that one site lies about 100 miles south-west of the capital, Addis Ababa, and the other another 100 miles or so to the north-east. Within each survey site, households reside in close proximity. Unfortunately, GIS tracking data were not included when the sample was collected and the survey designed, in 1997. However, it is not unreasonable to assume that the households residing within each respective 'peasant association' ('ilkura' and 'mirti', respectively) are likely to act as two intra-dependent but inter-independent 'neighborhoods'.

In order to incorporate the spatio-geographic information available, we assemble a known weight matrix (following protocols outlined in [61,62]; and, more recently, [63]). We then implement the covariate-selection and censoring-point investigation for a third time, allowing for a positively correlated spatial autoregressive Tobit regression. We permit the correlation across the known weight matrix to take *assigned values* in increments 0.00, 0.01, 0.02, . . . , 0.98, 0.99 (100 evaluations in total). Applying the [59] covariate-selection procedure and the censoring comparisons enacted above we execute further investigation about the likelihood of spatial correlation.

Once again, the results of the model-comparisons exercise evidence non-negligible correlations across households. The conjecture that spatial interactions between households within each of the two locations are important is confirmed. There appear to exist small but significant correlations across household sales activities within the random-covariate, zero-censored regression, and slightly larger correlations in the random-covariate, random-censored formulation. Thus, analysis should proceed with due regard for appropriate geo-spatial household interactions. We present evidence supporting this conclusion, subsequently.

### 4.4. Search Summary

Results of the model selection exercise are summarized in Figures 1–5. Figure 1 presents the search probability summaries for the covariate-design, threshold-design, and neighborhood-design experiments. Figure 2 reports the covariate probabilities derived from the random-covariate [59] algorithm. Figure 3 presents the marginalized evidence for the spatial autocorrelation parameter under the assumption that 'neighborhood effects' are present. Figure 4 presents the predictive measures derived from two models; one model is the 'preferred model', as determined by the full, comprehensive search procedure; and the other model is the 'conventional model', supplied previously in an initial version of this manuscript. Figure 5 presents the span of the log-marginal likelihood values over the entire model space; identifies where, in fact, the previously enacted ad hoc selection resides; and compares results with a random-covariate, random-censor, random-correlation search algorithm. The search algorithm demonstrates unprecedented, rapid convergence to the high-density space and raises considerable scope for additional application.

**Probability**

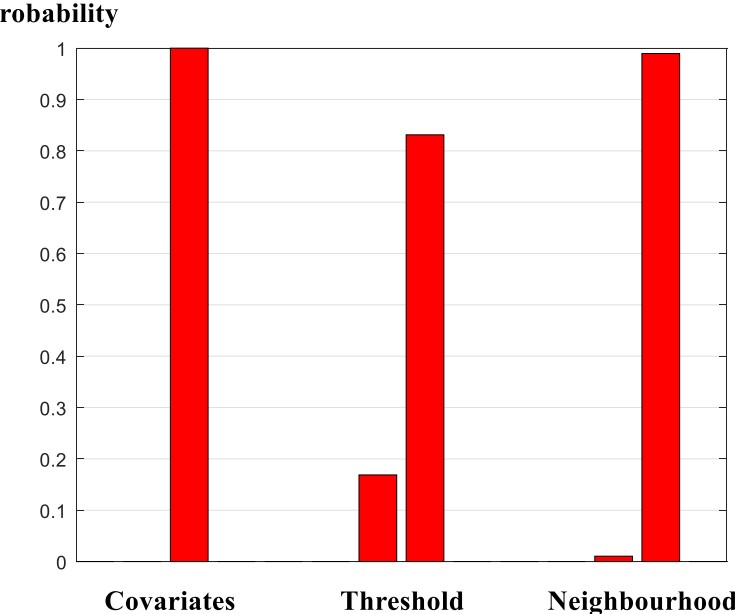

**Figure 1.** Model search probabilities for 'covariate design (left-most two entries)', 'censoring threshold (middle two entries)', and 'spatial autocorrelation neighborhood effects (right-most two entries)'. Reports for covariate-design experiment are, from left to right, the marginal probability for the seven-covariate assignment (as applied in previous works by the author) (0.000); and the marginal probability of random covariate assignment (1.000). Reports for the censoring point experiment are, from left to right, the marginal probability in favor of (conventional) zero censoring (0.170) and the marginal probability in favor of (non-conventional) random censoring (0.830). Reports for the spatial autocorrelation neighborhood experiment are, from left to right, the marginal probability in favor of zero neighborhood effects (0.001) and the marginal probability in favor of non-negligible neighborhood effects (0.999). There is substantial evidence in favor of the random-covariate selection design, with either conventional zero censoring or non-conventional random censoring and incorporation of neighborhood effects. There is virtually no support for the conventional framework applied previously by the author and supplied in a previous iteration of this manuscript. Noteworthy is the fact that the conventional Tobit specification presented in a previous round of this exercise is never once sampled within the 40,001-iteration selection exercise.

Viewed collectively, the model-selection exercise provides strong support for the random-covariate formulation (Figures 1 and 2); provides non-negligible support for both the random- and zero-censored threshold frameworks (Figure 1); and provides very strong support for the existence of neighborhood effects, within each geographical sub-sample, with positive spatial auto-correlation between neighboring supply activities (Figures 1 and 3). We conclude, therefore, that the evidence for the stylized, arbitrarily formulated specification, applied previously is, unfortunately, or, otherwise, depending on one's viewpoint, extremely low. We proceed accordingly, with due regard for the model-averaging implied by the results of the design search. In this context, it is useful to reconsider just how the focus upon a single, simplified framework may affect inferences drawn from the modelling exercise. Traditional posterior predictive conclusions (Figure 4) suggest that the posterior predictive capabilities of the traditional, *ad-hoc* specification may not be too poor and, in fact, dominate the alternative formulation. Noteworthy in this regard is that the ranking of these two specifications under an altogether separate criterion, is reversed (Figure 5). However, this observation belies a substantial and key component in the model-evaluation exercise: Whereas the former comparison (in Figure 4) uses point estimates in order to form conclusions; the current comparison (in Figure 5) accounts fully for the probabilities attached to the point predictions. Put another way, whereas the computations in Figure 4 rest on 'snap-shots' of the available predictive performance of the alternative specifications,

the computations underlying the plots in Figure 5 span the full, integral, probability space, placing appropriate 'weight' on each of the respective models comprising that specific 'space'. We proceed, accordingly, with due regard for the relative weights in evaluating the evidence of each of the models comprising the model space.

**Probability**

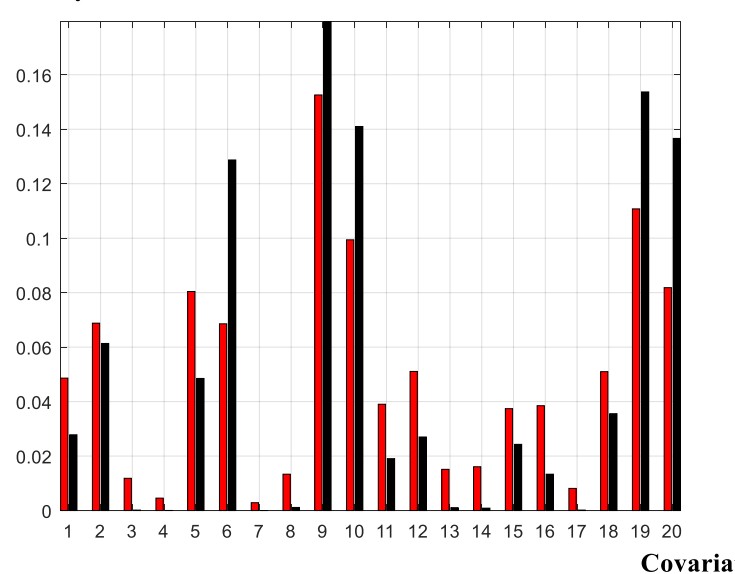

**Covariate**

**Figure 2.** Covariate probabilities derived from the random covariate-assignment scheme. The red bars denote 'probabilities' as measured by the 'inclusion frequency' within the Gibbs sample and black bars denote 'probabilities' using the marginal likelihood values derived from the Gibbs sample. The covariates are, respectively, from left to right: #1: 'gender'; #2: 'transportproxy'; #3: 'roaddistance'; #4: 'marketdistance'; #5: 'milkgroupdistance'; #6: 'credituse'; #7: 'experience'; #8: 'education'; #9: 'timetothemilkgroup'; #10: 'timewaitingatthemilkgroup'; #11: 'ilukura'; #12. 'mirti'; #13: 'cropland'; #14: 'pastureland'; #15: 'ownfarmequipment'; #16: 'owntransportequipment'; #17: 'extension'; #18 'localbreed'; #19: 'crossbreed'; and #20: 'constant'. Noteworthy are two observations. First, there is extremely low probability assigned to extension visitation (covariate #17), which was the focus of a previous iteration of this exercise. Second, there is relatively high probability assigned to 'time traversing to the milk group' (covariate #9), which is one of the principal transactions costs conjectures of the exercise, guiding, and directing data collection (Nicholson, 1997).

*4.5. Search Formalities*

Finally, with the interests of (a possibly diverse set of land-use experts, perhaps, more interested in implications, rather than technicalities and) pedagogy at heart; we have avoided formalism in the presentation of the results, used to generate the graphics in Figures 1–5. Readers wishing to apply the exercise may do so by executing a set of MATLAB© computer code protocols outlined in the appendix to this manuscript and made available by request to the author. Readers interested in technicalities will note that we have studied models with mathematical structure but avoided presenting their mathematical structures. In the presently revised setting, we make use of the previously enacted conventional framework, using it as one important benchmark, against which to consider 'improvement'. The benchmark framework referencing comparison and the alternative sets of frameworks can be considered and neatly assessed with reference to the data-generating exercise surrounding the 'i' = 1, 2, . . . , N = 68 households; the conventional Tobit, censored regression ([43]), where $z_i = x_i'\beta + u_i$, in which $z_i$ denotes a latent supply quantity (presently, the average supply of liquid milk per household per day during the production year); $x_i \equiv (x_{i1}, x_{i2}, \ldots, x_{iK})'$ denotes a K-vector of covariate quantities deemed essential to the supply decision; $\beta \equiv (\beta_1, \beta_2, \ldots, \beta_K)'$

denotes the K-vector of corresponding coefficients; $u_i$ denotes a random shock; and for 'i' = 1, 2, . . . , N, we observe $x_i$ and $y_i \equiv \max\{z_i, \tau_i\}$, wherein $\tau_i$ denotes a 'threshold'.

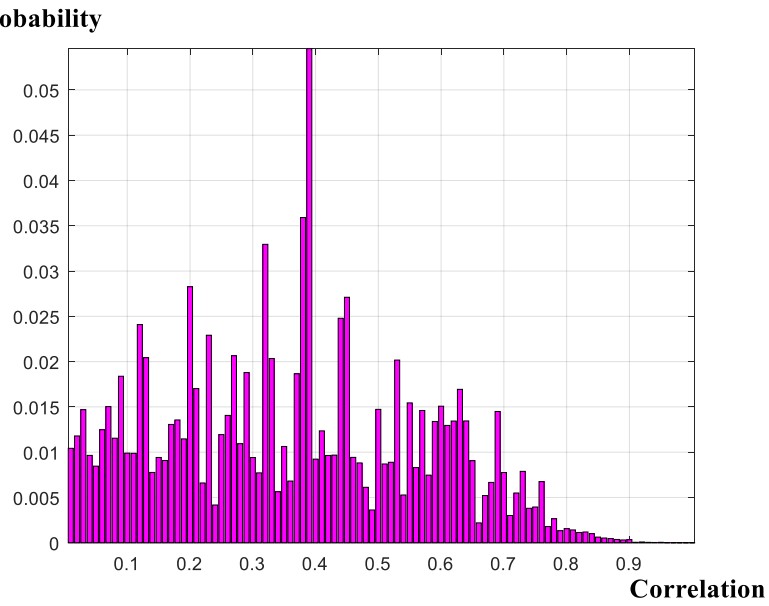

**Figure 3.** Model search probabilities for appropriate 'neighborhood design'. There is some support for the zero-correlation (spatial-independence, zero-social-interactions) assumption, but the evidence is small (0.01 compared to 0.99). Noteworthy is the 'spike' (mass point 0.0546), which occurs at the spatial autocorrelation value of 0.38. There is limited support for the conventional framework applied previously by the author and supplied in a previous iteration of this manuscript.

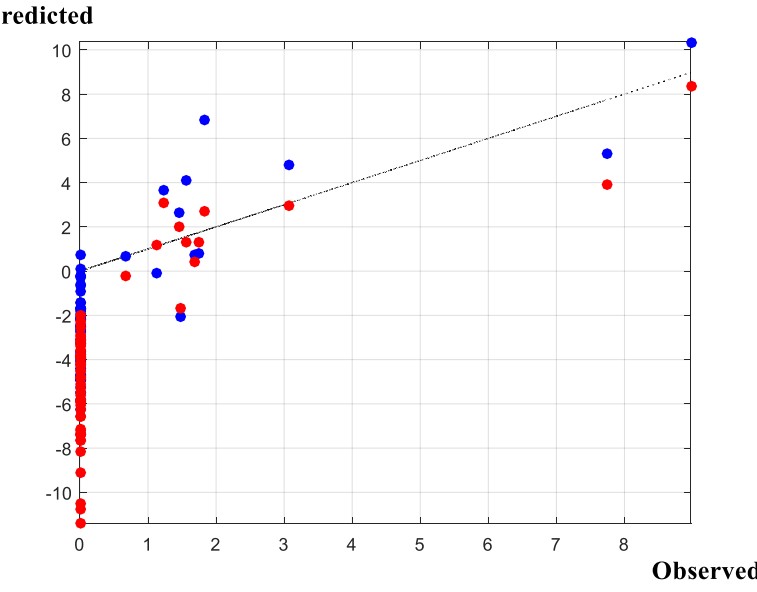

**Figure 4.** Predictive performances of two, respective, models. The black dotted entry depicts the 'line of perfect fit'; the red dots denote predictions derived from the 'preferred specification' and the blue dots denote predictions derived from the 'conventional specification'. There are 40,000 models evaluated in the model-search exercise. The 'model-averaged, preferred specification' uses the highest 100 posterior density estimates to form the weighted average. The two models appear to present very similar predictive capabilities. The 'conventional specification' predicts 56 of the 56 censored observations whereas the 'preferred specification' predicts 58; and the coefficients of determination between the observed and predicted reports for the 'conventional' and 'preferred' specifications are $(0.74)^2$ and

$(0.71)^2$, respectively. However, the evidence in favor of the 'preferred specification' completely dominates that of the conventional specification, with the posterior odds in favor of the former being substantial. The odds are $\exp(-47.69) \div \exp(-74.29) = 354{,}972{,}121{,}578.13$. Thus, whereas the odds in favor of the 'preferred specification' are substantial and completely overwhelm that of the 'conventional specification', the (superficial) predictive performances of the two specifications are comparable.

**Log Marginal Likelihood**

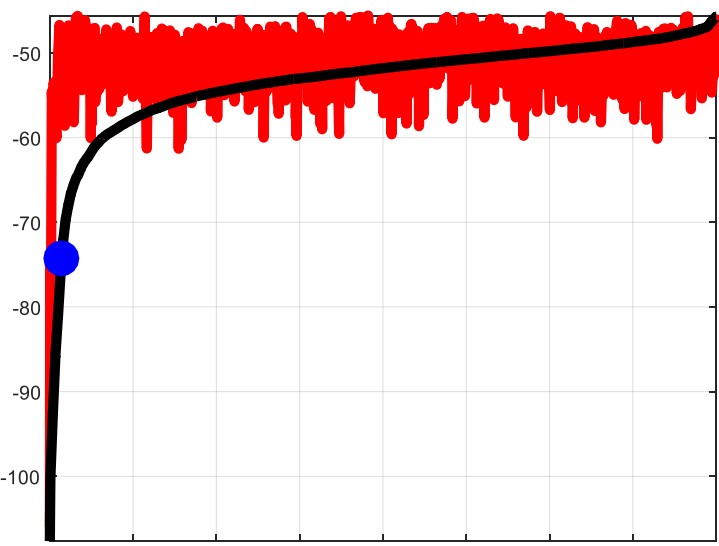

**Iteration**

**Figure 5.** Log marginal likelihood values derived from three sources. The black line depicts the pattern of log-marginal likelihood values derived from the full model search. There are 40,001 iterations in total. The values have been sorted in ascending order. The blue marker denotes the log-marginal likelihood value derived from the 'conventional specification' using zero censoring, zero spatial autocorrelation, and the seven covariates used previously in an initial iteration of this manuscript. This model was never accepted when sampled as part of the full model search. The third entries in red depict the pattern of log-marginal likelihood values derived from the algorithm, which searches randomly across the covariates, randomly across the 100-increment spatial-correlation index, and searches randomly across zero- and non-zero censoring points. Four features are noteworthy. The log-marginal likelihood value assigned to the ad-hoc conventional Tobit regression ignores significant mass between its value ($-74.27$) and the maximum available from the full search procedure ($-45.72$). Second, although the conventional Tobit regression log-marginal likelihood value significantly exceeds the minimum from the search ($-107.66$), it ranks only 39,292nd within the 40,001-unit model domain. Third, the random-covariate, random-autocorrelation, random-censor algorithm spans the model space entirely and compares well with the non-random search algorithm over the entire 40,001-unit domain, with a minimum value of $-105.92$ and a maximum value of $-45.58$. Fourth, and, perhaps, most significantly, the random-search algorithm quickly engages the high-density model space. The first iteration at which the log-marginal likelihood value exceeds that derived from the 40,001-unit search domain is at iteration #1575. Experiments with different starting values validate the conclusion that the random-covariate, random-censor, random-correlation algorithm is robust and evidences unprecedented, rapid convergence.

The notation is useful in the extent to which it forces us to consider 'fact' and 'fiction' and the sets of 'facts and fictions' the investigation is willing to 'maintain' or, otherwise, 'evaluate'.

Using the benchmark framework of the initial investigation, $\mathbf{x_i} \equiv (x_{i1}, x_{i2}, \ldots, x_{i7})'$ is arbitrarily designed to include the aforementioned seven covariates; the threshold, $\tau_i$, is set equal to zero; and the unobserved $u_i$ is assumed to follow a normal distribution with zero mean and variance given by the

square of another, unobserved parameter, $\sigma$. The unknowns are, then, $\theta \equiv (\sigma, \beta', \mathbf{z}')$, where $\mathbf{z} \equiv (z_1, z_2, \ldots, z_{No})'$ comprises the censored observations, and we have rearranged the ordering of the data so that the first $N_0 (<N)$ components reside foremost within the sample.

In order to compare 'fact' and 'fiction', recall [40]'s (1937) identity, $f(\mathbf{y}) \equiv \int f(\mathbf{y}|\theta) \times f(\theta) \, d\theta$, and note that the observed data $\mathbf{y} \equiv (0, 0, \ldots, 0, y_{No+1}, y_{No+2}, \ldots, y_N)'$ are all we have as 'fact'; that the covariates, comprising the (N × K) design matrix $\mathbf{X} \equiv (\mathbf{x_1}, \mathbf{x_2}, \ldots, \mathbf{x_N})'$, $\mathbf{x_1} \equiv (x_{11}, x_{12}, \ldots, x_{1K})'$, $\mathbf{x_2} \equiv (x_{21}, x_{22}, \ldots, x_{2K})'$, $\ldots$, $\mathbf{x_N} \equiv (x_{N1}, x_{N2}, \ldots, x_{NK})'$, while observable, are, of course, part of the 'fiction' because they need not correspond *at all* to the observed '$\mathbf{y}$', more than merely by assumption; note, in addition, that the threshold, itself, is 'fiction'; as is the assumption that the components of the (N × 1) vector of unobserved disturbances $\mu \equiv (\mu_1, \mu_2, \ldots, \mu_N)'$ have corresponding normal distributions; that the random shocks are, indeed, independent, implying that the vector of disturbance terms, $\mathbf{u}$, has the normal distribution $N(\mathbf{0_N}, \sigma^2 \mathbf{I_N})$; or that, *possibly*, but, more likely, *probably*, there are other assumptions (perhaps very many) that we have made implicitly or, rather, unknowingly, leading to the standard censored Tobit regression ([43]), $\mathbf{z} = \mathbf{X}\beta + \mu$, observing $\mathbf{y} \equiv \max\{\mathbf{z}, \mathbf{0_N}\}$, where the N-vector $\mathbf{y}$ is produced from row-wise comparisons between the N-vector $\mathbf{z}$ and the N-vector $\mathbf{0_N}$.

The notation helps also to make explicit the three categories of status within which we find ourselves, namely observed data, $\mathbf{X}$ and $\mathbf{y}$; modifications to the set of covariates embedded within $\mathbf{X}$; and assumptions about the threshold levels, $\tau$, and the presence or absence of social interactions, governed by correlation, which we now explicate. We consider model search in the context of

$$\mathbf{A} \, \mathbf{z} = \mathbf{X}\beta + \mathbf{u}, \tag{1}$$

observing $\mathbf{y} \equiv \max\{\mathbf{z}, \iota_N \tau\}$, where (the N × N) matrix $\mathbf{A} \equiv \mathbf{I_N} - \rho\mathbf{W}$, reflects spatial dependence; $\mathbf{I_N}$ denotes the N-dimensional identity matrix; the scalar '$\rho$' depicts the magnitude of spatial autocorrelation; (the N × N) matrix $\mathbf{W}$ denotes the 'pattern' of (given) neighborhood effects at the two, respective survey sites; the scalar '$\tau$' denotes the threshold; and $\iota_N$ denotes the N-dimensional unit vector. We make use of (1) subsequently (considering it obviously succinct but perhaps also more directly informative), noting that the structure ensues from the data generating mechanism, $z_i = \rho \sum_{j \neq i} w_{ij} z_j + \mathbf{x_i} \beta + u_i$, where '$\rho$' denotes the particular form of neighborhood correlation evolving between members of the 'neighborhoods' which are defined by $w_{ij}^{(s)}$, 'i' = 1, 2, $\ldots$, N, 'j' = 1, 2, $\ldots$, N, such that, under the circumstance that $w_{ij} = 1$, then households 'i' and 'j' ('i' ≠ 'j') are considered 'neighbors' whereas $w_{ij} = 0$ denotes otherwise. Finally, in terms of implementation, and with the notation surrounding (1), at hand, the search algorithm proceeds by considering 'proposals' $\mathbf{X}^{(p)}$, $\tau^{(p)}$, and $\rho^{(p)}$, and setting the respective proposals as 'states' $\mathbf{X}^{(s)}$, $\tau^{(s)}$, and $\rho^{(s)}$ with probability $\min\{f(\mathbf{y}|\Delta^{(p)}) \div f(\mathbf{y}|\Delta^{(s)}\}$ where $\Delta^{(s)} \equiv \{\mathbf{X}^{(s)}, \tau^{(s)}, \rho^{(s)}\}$, $\Delta^{(p)} \equiv \{\mathbf{X}^{(p)}, \tau^{(p)}, \rho^{(p)}\}$ and at least *one* of the conditions $\mathbf{X}^{(p)} \neq \mathbf{X}^{(s)}$, $\tau^{(p)} \neq \tau^{(s)}$ or $\rho^{(p)} \neq \rho^{(s)}$ exists. Again, the algorithm is an adaptation of the ideas presented seminally from within [59] and embedded within MATLAB© computer code, itemized within the appendix, and made available upon request. We remind the reader that, although the actual correlation value is unknown, the 100-increment assessment would result in exact computations were the censored values for the sales quantity also known. The censoring of the sales-dependent variable leads to a well-known problem of evaluating cumulants of a truncated-multi-variate-T distribution (see [64,65]), which arises within the Tobit marginal likelihood and is problematic. However, modification of the Geweke–Hajivassiliou–Keane algorithm ([66–71]) makes evaluation of the Tobit, censored-regression, marginal likelihood routine, efficient and robust.

The findings presented here raise scope for additional empirical enquiry.

## 5. Distance-to-Market Estimates

The predictive results presented in Figure 5 confirm that both the model-averaged and conventional Tobit formulations present 'satisfactory representations of the data generating process'; and the model selection results, in Figures 1–4, suggest that adherence to a random threshold and to incorporation

of spatial neighborhood effects is advisable. Here we redirect attention in three directions, toward strategies that might be appropriate as market-precipitating catalysts, namely increasing numbers of cross-breed cows, increasing numbers of local-breed cows, and increasing extension visits to each of the non-participating households.

Concerning the costs and benefits of the separate interventions, each of the strategies produces the same benefit (a given number of new entrants) but outlays different costs, where each cost depends on the total amounts of resources required to achieve the objective and the per-unit costs of each of the separate resources. We derive estimates of the resource requirements from the Tobit regression. With reference to Equation (1), by accounting appropriately for the spatial geographic matrix $\mathbf{A}$; subsequently, setting $\mathbf{z}$ equal to the censoring threshold; we can compute the levels of resources required for non-participants to participate. Formally, for each household within the set of censored observations $\mathbf{c} = \{i|y_i = 0\}$, the level of some resource, say, 'resource k', required to effect entry is represented by the vector

$$\hat{\mathbf{x}}_{\mathbf{k}} = \frac{\mathbf{A}\,\tau\,\iota_{\mathbf{N}} \;-\; \mathbf{X}_{-\mathbf{k}}\beta_{-\mathbf{k}} - \mathbf{u}}{\beta_k} - \mathbf{x}_{\mathbf{k}} \qquad (2)$$

where $\hat{\mathbf{x}}_{\mathbf{k}}$ denotes the N-vector of additional resources required; $\mathbf{A}$, $\tau$ and $\iota_{\mathbf{N}}$ are previously defined; $\mathbf{X}_{-\mathbf{k}}$ denotes the (N × (K-1)) covariate matrix obtained by removing the kth component; the coefficient-vector $\beta_{-\mathbf{k}}$ denotes the (K-1)-vector of corresponding coefficients; $\beta_k$ denotes the (scalar) amount by which sales change when we change the resource in question; and $\mathbf{x}_{\mathbf{k}}$ denotes the household's present endowment of the scarce resource (As noted by an anonymous referee, the quantity, '$\mathbf{u}$', appears within the numerator within Equation (2) due to the primal definition of the Tobit regression within Equation (1). However, because its expectation is, of course $\mathbf{0_N}$, we suppress it from entering the MCMC calculations used to estimate the distance estimates). In short, household 'distance estimates' are easily retrieved from a transformation of the Tobit regression.

One problem arising in comparisons including 'extension' is the fact (Figure 2) that this particular covariate has an extremely low occurrence level within the covariate-selection exercise. Figure 2 suggests that this measure has, at best, a chance of only about one percent of occurring within any single Gibbs sequence. In contrast, local-breed cows and cross-breed cows are highly probable causal factors affecting milk sales and milk-market participation, correspondingly. A previous version of this paper considers 'extension' as a market precipitator; however, now, with the advantage of formal model comparisons at hand, this assumption appears to be erroneous. In what follows, we confine attentions to the two cow adoption strategies; on the one hand, expansion of the holding of indigenous-breed animals and, on the other hand, expansion of exotic-breed animals.

Using Equation (2) we are able to generate distance estimates for each animal breed and for each non-participant household within the sample. Further, we are able to repeat the exercise making comparisons about the predictive differentials eschewed by the alternative models, namely the previously applied 'conventional Tobit regression' and the 'model-averaged, preferred specification' as determined by the model-selection exercise.

Figures 6 and 7 report the distance estimates derived from the Gibbs sample. Across the two graphics there are, as expected, some significant differences about the total numbers of local-breed and cross-breed cows deemed essential for market entry. However, within each graphic, there appears some considerable 'likeness' in the reports derived under the preferred, model-averaged specification, and those of the previously applied, conventional specification. Thus, continuing a thematic originating from within Figure 4; at least as far as point estimates are concerned, there does not appear to be much that is injurious in applying the simplified Tobit censored regression framework in order to obtain distance estimates.

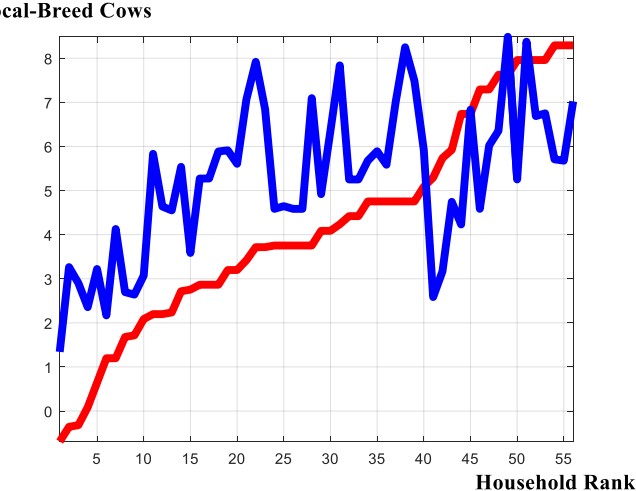

**Figure 6.** Local-breed cow distance estimates. The red line reports the model-averaged distance estimates resulting from the preferred specification. The households have been ranked in ascending order of their 'requirements' such that the household deemed 'nearest' to the market has an 'excess' of, approximately, one cow, whereas the household deemed 'farthest' from the market has a 'deficit' of more than eight local-breed animals. The blue line reports the distance estimates derived from the conventional regression, retaining the household rank derived under the preferred specification. Noteworthy are two features. First, both series indicate a similar overall amount of 'deficiency' across the non-participating household sample; the preferred-specification reports rise monotonically from a minimum of −0.69 to a maximum of 8.29, with a mean of 4.21 and a median of 3.92; the conventional-specification reports rise from a minimum of 1.34 to a maximum of 8.34 with a mean of 5.27 and a median of 5.41. Second, despite the overall similarity in estimates, there are substantial differences when the estimates are compared across households. The household in which the difference between the series is smallest (a difference of 0.01) is household #45 and the household within which the difference is largest (a difference of 4.20) is household #22. Thus, although the evidence in favor of the preferred specification vastly outweighs the evidence in favor of the conventional specification, the distance estimates reports for local-breed cow deficiency appear to be little affected.

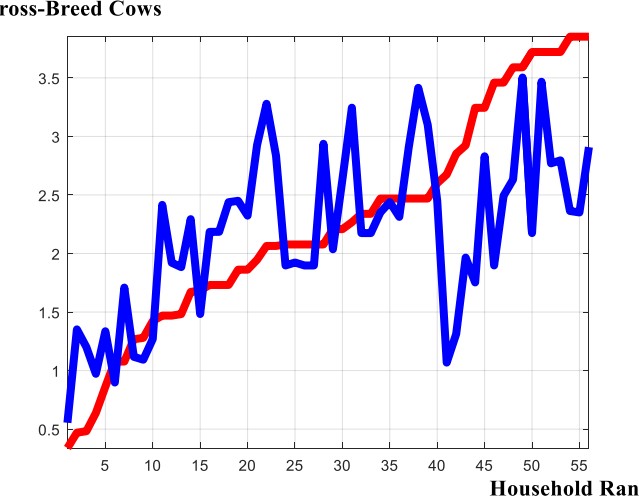

**Figure 7.** Cross-breed-cow distance estimates. The red line reports the model-averaged distance estimates resulting from the preferred specification. The households have been ranked in ascending order of their 'requirements' such that the household deemed 'nearest' to the market has a 'deficit' of, approximately, one cow, whereas the household deemed 'farthest' from the market has a 'deficit' of more than three cross-breed animals. The blue line reports the distance estimates derived from the conventional regression, retaining the household rank derived under the preferred specification. Noteworthy

are two features. First, both series indicate a similar overall amount of 'deficiency' across the non-participating household sample; the preferred-specification reports rise monotonically from a minimum of 0.34 to a maximum of 3.85, with a mean of 2.26 and a median of 2.14; the conventional-specification reports rise from a minimum of 0.55 to a maximum of 3.50 with a mean of 2.18 and a median of 2.24. Second, despite the overall similarity in estimates, there are substantial differences when the estimates are compared across households. The household in which the difference between the series is smallest (a difference of 0.03) is household #35 and the household within which the difference is largest (a difference of 1.61) is household #41. Thus, similar patterns are observed across the two sets of series. We note additionally, two features when Figures 6 and 7 are viewed collectively. First, the inter-specification reports are almost identical in pattern, with correlation coefficients between the local-breed and cross-bred cow deficiencies almost 1.00 under the preferred specification and, again, almost 1.00 under the conventional specification. The intra-specification reports are also highly correlated; the correlation of coefficients between the two local-breed cow reports is 0.59 and between the two cross-breed-cow reports is also 0.59.

## 6. Land-Use Pathway Ranking and Selection

Implementing the preceding graphics in a cost–benefit framework requires information about the likely costs of the various policies. In the cases of the two strategies under consideration the costs are mostly private. In particular, the costs confronting the households are simply the economic cost of the purchase of capital stock. However, and by way of further motivation of land-use pathway implementation, the pathway chosen may have immensely different *social costs*, in addition to the potential benefits accruing, privately, to each individual household. In particular, there may be substantial costs emanating from 'externalities' which we are unable to consider adequately within this exercise. For example, erosion concerns within the Ethiopian highlands are recurrent and the literature on 'pastoralism' and 'agri-environmental intervention' is replete with examples connecting stocking rates and pastoral degradation. Given these concerns, it is likely that the choice between cross-breed and local-breed livestock units affects environmental concerns. While there is some comfort in noting that the impacts of the alternative strategies may be very close, there are, likely, a number of alternative factors that can influence costs, and we proceed with this caution in mind.

In order to impute relative costs differentials, we exploit the household-production model ([26]), cash-income constraint, namely.

$$v \le pf(\mathbf{x}) - py_c - \mathbf{wx} + \omega \tag{3}$$

where the quantity 'v' denotes a (numeraire) good purchased in the market-place; parameter 'p' denotes the per-unit price of the agricultural product; the function $f(\cdot)$ denotes the milk production technological possibilities available to the household; the quantity '$y_c$' denotes the household's level of milk consumption; the vector $\mathbf{w} \equiv (w_1, w_2, \ldots, w_m)'$ denotes the per-unit costs of (purchased) inputs that are required to produce milk; $\mathbf{x} \equiv (x_1, x_2, \ldots, x_m)'$ denotes the corresponding variable input vector; and the quantity signified by parameter '$\omega$' denotes exogenous, household income. The household-production model, cash-income constraint, produces the marginal productivity relations.

$$f_k(\mathbf{x}) = w_k \div p, \text{ 'k' } = 1, 2, \ldots, m. \tag{4}$$

These marginal productivity relations provide bases for standardizing the costs comparisons and identifying 'best' policy. Up to a normalizing constant (the per-unit market-price of milk, 'p') the marginal products give us cost imputations enabling cost–benefit evaluations of the alternative policies. Of course, in the event that actual, per-unit, livestock values are available, these values can be used directly, instead of the marginal-productivity estimates. However, and perhaps, surprisingly, these values were not recorded and are likely to differ considerably across the respective study sites. The marginal-productivity values are, however, retrievable from the data supplied within the survey. Another alternative, as suggested by an anonymous referee, is simply to use, directly, the estimates

evolving from the separate Tobit regressions. Indeed, this procedure is valid in the event that the livestock units do not affect consumption, which is tenable.

Using the household-production, cash-income constraint, one could apply the Tobit censored regression estimates for cross-breed and local-breed units, directly. However, [45,49] originally estimated a linear relationship between output (milk production) and the inputs (cross-breed cows and local-breed cows) using the data applied in the present investigation and presented within an earlier iteration of this exercise. The marginal products of cross-breed cows and local-breed cows (with associated 95% highest-posterior density intervals) are, respectively, 3.02 (2.72, 3.26) and 1.24 (1.02, 1.40). Considerable uncertainty exists about the precise value of the marginal-product estimates one should employ. The values applied presently are the values we have applied previously in an initial iteration of this exercise. We retain these same estimates, observing that these estimates, too, should be formalized and incorporated as part of the model-selection exercise. However, experience suggests that the marginal product point estimates are relatively stable under a diverse array of alternative specifications. Using the ratio 3.02:1.24; scaling the probability distributions underlying the distance estimates; iterating the Tobit estimation a number of times; and computing, within each iteration, the least-cost strategy is depicted through the probability distribution reports appearing in Figure 8.

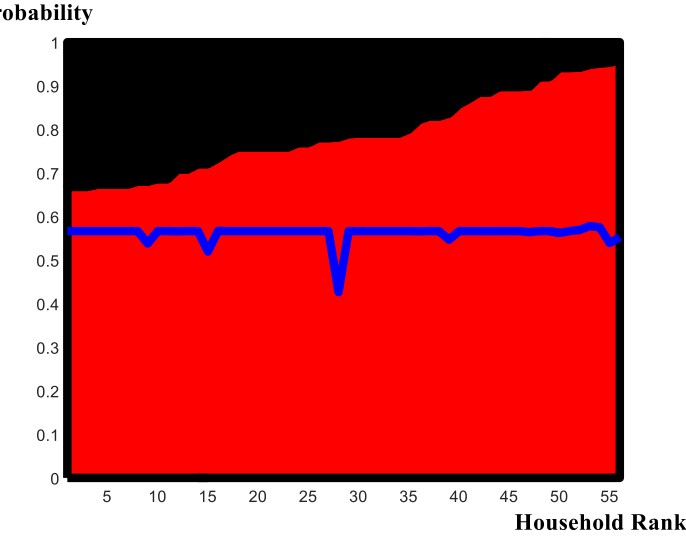

**Figure 8.** Ranking and selection probabilities across the non-participating household sample. The red area depicts the probability that 'local-breed cow adoption' is the cost-efficient strategy based on the preferred specification. The black shaded area depicts the probability associated with the 'cross-breed cow adoption' strategy based on the preferred specification. The red and black shaded areas have been sorted in ascending order of the 'local-breed cow adoption strategy' (descending order of the 'cross-breed cow adoption strategy'). The blue line depicts the break-point between the local-breed-versus-cross-breed cow adoption strategy derived from the conventional Tobit censored regression specification. The reports are, once again, sorted on the preferred specification local-breed cow adoption strategy. Noteworthy are two features. One is the lack of ascension in the conventional-Tobit specification compared with the preferred-specification reports. Second is the overall lower probability assigned to the cross-breed cow adoption strategy. In conclusion, concerning predictions about ranking and selection, there appear to be significant differences propagated by the two, respective specifications.

In Figure 8, the solid shaded area in red depicts the probability that the local-breed cow strategy is 'best', as obtained from the 'preferred specification'. The solid shaded area in black depicts the probability that the cross-breed cow strategy is 'best'. The reports have been ordered in ascending value of the local-breed cow strategy and descending value of the cross-breed cow strategy. The overwhelming support appears to uphold the notion that the 'better' of the two approaches to effect market entry among the non-participating households in the Ethiopian sample is the strategy of expanding ownership

of local-breed animals. Notwithstanding this remark, there is non-negligible support for the alternative cross-breed cow strategy. In the reports we have, once again, super-imposed the estimated probability 'breakpoint' between the local-breed and cross-breed strategies derived from the conventional Tobit censored regression specification. This break-point sequence is ordered using the same index of the households (as determined by the ascending local-breed cow probabilities under the preferred specification). Some significantly different patterns emerge between the two reports, and so, unlike the predictive reports obtained from the Tobit regressions and the 'distance point estimates' (Figure 4, Figure 6, and Figure 7, respectively), the ranking-and selection outcomes appear significantly affected by choice of model specification.

The results suggest that, in 100% of all possible cases, the local-breed cow expansion strategy dominates. Although the cross-breed cow strategy has non-negligible support, it is never a dominant strategy within any single household.

The results suggest our main conclusion, which is that, considering available alternatives, the policy of increasing indigenous-breed animals is best.

## 7. Land-Use Policy Applications and Extensions

We now consider (in addition to the methodological aspects, scrutinized, above) several *practical* aspects of the estimation procedure, together with a few, additional, examples within which the over-arching ideas, and some possible extensions of them, may prove 'fruitful'. For this purpose, it is useful to reflect on just what we have accomplished thus far.

### 7.1. Primary Contributions

Although it may potentially mitigate the value of its contribution, we have demonstrated the *simplicity* with which formal model comparisons can be executed in the Tobit censored regression framework when the investigator has substantial model uncertainty. We have emphasized the highly important aspect of model search in the covariate domain; we have emphasized the importance of search within the geo-spatial-autoregressive domain; we have emphasized search's importance within the threshold domain; and we have (perhaps more importantly, at least from a methodological perspective) indicated just how such search can be enacted, executed, and interpreted with reference to relatively minor adaptations of one very powerful algorithm [59].

We have also highlighted the tractability of the Tobit regression structure as a *modus operandi* for studying market participation, but also (although somewhat de-emphasized) comparing policy interventions.

Methodologically, we have succeeded in extending the basic Geweke–Hajivassiliou–Keane algorithm ([66–71]) for estimating the cumulants of the truncated multivariate-T distribution that arise when evaluating the Tobit marginal likelihood ([64,65]).

Additionally, we have taken lengths to assess the practical implications of the model-search conclusions and, especially, the sensitivity of another, albeit, conventional procedure (as supplied within a previous version of this manuscript). We conclude that, as far as the Ethiopian-highlands sample is concerned, and in the context of the spatial-autoregressive, random-censored, random-covariate Tobit regression, these comparisons can be performed fairly routinely using refinements of standard *Monte Carlo* procedure. We have demonstrated substantial support for the model-averaged specification. We have indicated several situations in which the adoption of a more conventional approach is not too injurious. We have also indicated one especially important setting in which the neglect of model search can greatly affect conclusions. These fairly heterogeneous frameworks allow the investigator to have an informed view about the robustness of the conclusion that one particular policy is preferred to another, or, possibly, via extension of the bipartite application presently studied (herd expansion of either local-breed, grade indigenous stock versus herd expansion of cross-breed exotic, imported stock) to multitudinous comparisons across more than two covariates.

Over-arching the inferences derived within the alternative regimes is the important question about how much 'weight' the data place upon the competing models under investigation and, despite its significance, it is important that the reader understands that 'all' we have done is repeat this computation for a variety of alternative setups. This simple procedure is informative; and, by way of example (Figure 5) relegates an *ad hoc*, subjective choice to a relatively lowly 'rank' in the overall, large-dimensional model space. We have shown that in at least one very important setting—namely, a choice across livestock type (Figures 6–8)—that allowing appropriately for the 'weight' assigned to different econometric specifications can greatly affect conclusions.

Additionally, and, depending on the reader's viewpoint, one 'negative', or, possibly, 'positive' finding evolves from the covariate selection exercise. We indicate that it is likely that conclusions drawn in previous applications of the data could possibly be overturned, or, at least, that the marginal-likelihood weights (which those previous studies have not computed) likely relegate them to only 'casual' interest. Hence, non-invocation of relevant evidential material (the marginal-likelihood) may direct investigation toward false conclusions.

Finally, the model-selection exercise extends the extant ranking-and-selection procedure (enacted on a single model formulation) which, recall, was applied within 22].

Before turning to review related land-use contributions where we think that the 'new' methodology might be meritorious, we emphasize the simplicity with which these results are derived. The algorithm enabling model comparisons and facilitating formal probabilistic evaluations in the presence of substantial model uncertainty (Figure 5) is extremely straightforward to implement (MATLAB© companion appendix) and has the potential to facilitate additional comparisons in a potentially wide variety of diverse empirical settings. These settings, it must be emphasized, need not pertain exclusively to Tobit estimation [43]. These settings are considerably broader than the specific foci presented here. Also, these settings are availed to us by virtue of the single unifying quantity, $f(\mathbf{y})$; its significant, expanding, heritage and precision of computation ([40–42,72–75]); and the ultimate facility that the ready comparison of non-conjugate settings, such as the Tobit, are now availed to us. The basic work to be enacted in order to generalize from the Tobit censored regression framework involves only refinement of the single unifying measure, $f(\mathbf{y})$, and appropriate Monte Carlo implementation, namely.

$$f(\mathbf{y}) \;\cong\; \frac{1}{S}\sum_{s=1}^{S} f(\mathbf{y} \mid \mathbf{\Theta}^{(s)}) \tag{5}$$

This attractive feature of formal Bayesian analysis is something that we generally de-emphasize in empirical settings, but is, in the author's opinion, the *modus operandi* for expansive application in a wide and diverse set of land-use policy settings. We present a very brief 'helicopter tour' of some of these applications, subsequent to reflecting upon the major limitation of our study, as it affects, primarily, land-use pathway ranking and selection.

### 7.2. Primary Limitations

We have extended thematic developments concerning livestock choice to one embracing additional, significant features of the socio-economic setting of the Ethiopian highlands sample. To the extent that land-use decisions concerning livestock choice embrace concerns in agro-environmental contexts—contexts of over-arching empirical importance—we need to assess these, or at least, reflect on these, in the context of assessing conclusions. These contexts concern, in particular, the impacts of changes in stocking rates on the increased potential for soil erosion, the potentially deleterious effects that increased livestock intensities may engender on overall productivity of grasslands, and the sustainability of given stocking rates in the Ethiopian agro-pastoral complex.

Concerns in these directions were contemporaneous at the time data were collected (see, for example, [37], which is replete with stocking-rate contributions) but continue to this day ([76]). A significant heritage in stocking-rates debates exists in the general agricultural-science literature

(see, for one important example, [77]) but, especially, within the Ethiopian grasslands, highlands, farming-systems setting ([78–81]). These studies are significant in the context of our main results (Figure 8) prompting us to condition those findings on a number of significant impediments which could, possibly, overturn rankings. In additional to the minor methodological weaknesses of our empirical enquiry (neglect of the panel structure of the households, neglect of a hierarchical modelling intervention, and neglect of possibly other covariates, or refinements, such as permitting separate geo-spatial social interactions at the two respective sites), the main limitation engendered by this rich literature on stocking rates is our failure to account for environmental externalities and consequent refinement of the cost–benefit calculations so enacted. This one limitation is noteworthy and one which should ultimately stem re-direction in any future extension of the current effort. Another possible limitation that needs accounting is the presence of additional public costs implicit in any public intervention (see, for example, [82]); and these costs clearly need incorporation in any publicly provided intervention that might assist future market immersion. Other limitations, no doubt, exist, but, we conjecture, are likely less significant than the deleterious consequences of the alternative stocking rates across the markedly different animal units. It remains to be seen how such refinement could dramatically affect the conclusions derived in Figure 8.

### 7.3. Primary Land-Use Applications and Extensions

Thus, while at least one important opportunity for reanalysis exists with respect to the present data, some additional opportunities, elsewhere, may also arise, with alternative data and different, but related foci. In the context of considering (re-) application and extension of these ideas in related work, we embark on a very brief (and admittedly superficial) 'helicopter tour' of several noteworthy, previous contributions.

From within [83] the authors consider a national-scale analysis of the determinants of agricultural land prices in the contiguous United States. Their ideas are based on the conventional economic wisdom that the price of land, within a competitive market, should equate to the discounted sum of expected returns obtained by allocating the land resource to its most profitable use. Claiming that, in the event 'the most profitable use' might change at some future period, then the returns should reflect such change in a simple additive form with the net present value reflecting the income stream from the first use (in their example, agriculture) summed with the discounted stream from the subsequent new development (industrialization). The authors of [83] employ a classical, spatial-econometric model in order to decompose the (present) farmland price into its agricultural- and non-agricultural income-stream components. There is neither examination of potential model uncertainty nor comparison of the change-point analysis of when the developmental (from agricultural-to-non-agricultural) land-use adjustment occurs. Extant analysis, as robust as it appears, could potentially be enhanced by considering the 'best' time at which to make the agricultural-to-non-agricultural land use change, which, they emphasize, within their modelling context, is 'stochastic'. Conceivably, such analysis could draw upon an important early body of work ([84–86]) for which multiple change points arise and further extension, to consider the ranking and selection of the separate change points, using present analysis, may yield additional fruit.

In related work, [87] consider land-use change prediction from global, agro-economic model comparisons, arguing that large uncertainties about land-use demands for environmental services arise, principally, due to the high degree of model uncertainty embracing the various estimates. Using population, gross-domestic-product, and biophysical-yield changes in their simulations as potential drivers of change; they embark upon a comprehensive treatment of the model uncertainties pervading previous analysis. Focusing on two, primary research questions—namely, how much cropland will be used under different socioeconomic and climate-change scenarios and how can differences in model results be explained—they employ computable-general and partial-equilibrium structures in order to form conclusions. Hence, although model uncertainties appear at the forefront in the minds of the authors, there is no adherence to the work [40] embodying protocols availing formal model comparison.

The suitability of present methodology for comparing alternative policy simulations could be possible once appropriate data are presented. In this case, one envisages that it is possible to rank and select across the various land-use and climate-change scenarios. Using [87]'s welfare scores as the unifying ranking criterion, an analysis not unlike the present could evolve.

Recently, [88] considers changes in the market values and the geographic dispersion of changes in United Kingdom agricultural production under various climate-change and policy scenarios. The scenarios pertain, essentially, to the levels of emissions extant within the economy and the relative strengths of in situ environmental regulations. There is substantial model uncertainty inherent in analysis. There is substantial uncertainty inherent in the mathematical-programming exercise, most obviously in the values of particular coefficients used to 'generate solutions'. There is also inherent uncertainty in the regression interventions used to form predictions. Also, there is additional uncertainty arising within the 'marriage' of assumptions formed within the regression analysis with those embedded within the programming analysis. The presently piecemeal fashions in which assumptions are incorporated raise ameliorative scope for re-application of the single, unifying ([40]) quantity. Alternative policy options, the multitude of diverse assumptions underlying analysis, and the absence of formal model comparisons draw into question the validity of empirical findings. By targeting a single unifying exercise, and ranking and selecting across the policy options available, improvements may be possible. Whether the essential data are available for this purpose remains to be seen.

In an earlier paper, exemplifying the intense interest in the precise relationship between land-use and climate change, [89] measures the impact of land-use change on surface warming in the United States using data from 50 years prior to their analysis. The analysis includes regression, with simple interpolation, and linear relationships between covariates and the response variable (surface temperature). There is neither account for model uncertainty nor censoring for the latent observations. Thus, a direct extension of the data augmentation principal underlying Tobit, probit, and latent-variables estimation (see, for general discussion, [13–15,43,55,62,64–66], among others) could further cement main conclusions.

More recently, Ref. [90] reviews the current status and uncertainties surrounding shortcomings of existing models of land-use change and associated greenhouse-gas emissions as a result of bio-fuel production; evaluate options for 'model improvement'; and for conducting additional analysis. The authors identify many assumptions as key impediments in consistently evaluating the conclusions of indirect land-use change, in addition to market-based competitive-equilibrium assumptions. The authors highlight this need as 'fundamental' to the valid development of effective policy and sustained land-use intensification where bio-economic fuels generation is at issue. The paper is very similar in 'spirit' and in 'thematic development' to present intentions, with due regard for the sensitivities affecting land-use policy formation in the presence of substantial model uncertainty. Embellishment of the ideas presently espoused through a single unifying quantity, $f(\mathbf{y})$, could ameliorate remaining concerns.

More recently, within the *American Journal of Agricultural Economics*, and in a context intimately related to present efforts, [91] devotes attention to the problem of market participation in the Kenyan smallholder dairy sector. Arguing that conventional methodology fails to account adequately for market enticements to induce 'relevant non-producers' (producers 'residing' within a similar geographic locale, or producers of related produce with the possibility to 'mobilize' resources in order to take advantage of the newly arisen market opportunity) to become incumbent. They devise a so-called 'triple-hurdle model' of market participation in which an 'initial stage of production includes the status, 'non-production'. They use the model to identify the factors associated with Kenyan smallholders choosing to participate in dairy production and the role that these producers play, or not, within the marketplace. They produce a 'likelihood ratio test' in order to consider the extent to which their new methodology is improved by incorporating this new reference category, along with the 'third hurdle' in milk-market entry in their East-African sample setting. Their analysis is robust and makes the

analysis undertaken presently highly topical, emphasizing its 'similar vein of enquiry'. Their valuable, East-African milk data beckon re-application toward some of the technologies presented currently, and the sense that, perhaps, additional 'improvement' may be possible. Scope appears to exist for the possibility of fruitful re-examination of their East-African milk market data. The extension is argued to be possibly the single most direct extension of present efforts. More importantly, presently, the intriguing possibility of a neglected 'third hurdle' exists.

More recently, within *Journal of Environmental Economics and Management*, [92] considers the dynamics of indirect, land-use change focusing on empirical interventions within the Brazilian landscape. They expressly focus on the indirect effects of sugarcane expansion on the forest-conversion decisions in the country's Amazon region. They include a wide variety of models in their reported empirical exercise (not to mention, likely, others where space limitations preclude reporting) and make decisions based on an array (models one through seven, Table 1, p. 384) of alternative assumptions. These assumptions, it is noteworthy, are quite similar to the types of model-based uncertainties entertained presently. In this context, the availability of a central unifying quantity ([40]) makes neglected comparisons automatic.

In a recent *Land Use Policy* contribution, Ref. [93] delineates the determinants of households 'most affected' by deforestation in their fuelwood and non-timber forest-product collections in Cambodia. The instigating factor for the case study analysis is the view that tropical lowland forests are decreasing in size due, mainly, to agribusiness development and expansion of farming enterprise. Such areas, (as determined from satellite imagery) and related, arbitrarily chosen, 'susceptible districts' motivate household-data collection across a total of 161 respondents. Households were selected from within six villages emanating from within six geographic districts. The five-year sample evidences considerable diversity in 'land-use coverage' and in 'change-in-land-use coverage'. Using generalized-linear-regression techniques, they examine the principal determinants of household deforestation affectation. Nine covariates are examined (the deforestation area; the collection site; the main livelihood activities; experience of forest clearing; and the amount of exogenous, material wealth). 'Aikake's Information Criterion' is used to select 'best-fit' models for each fuel-wood and non-timber-forest-product collection specification. Conclusions are drawn after iterating investigation across a total of *ten* alternative specification forms. An ostensibly 'classical' version of (Bayesian) 'model-averaging' is enacted whereby conclusions about the strength of dependence of affected behavior on covariates is derived by 'averaging' across individual determinants using the proportions of Akaike-Information-Criteria for individual model interventions as weights. The spirit of enquiry is analogous to present efforts. The model-averaging procedure, covariate-selection exercise, and the ultimate conclusions drawn are relevant to the thematic developments entertained presently in an altogether Bayesian setting; and it would be very interesting, informative, and potentially insightful to reassess [93]'s data using present techniques.

Recently, in *Ecological Economics*, [94] investigates the effects of urban land use on residential well-being in major German cities, using panel data from the German Socio-Economic Panel, combining it with cross-sectional data from the European Urban Atlas. The independent variable 'satisfaction with life' is constructed from an 11-point Likert-scale that asks respondents "How satisfied are you with your life, all things considered?" A 'rich set of observables' amounting to some 40 correlates is included. Land-use categories of forests and water bodies are encompassed in order to reflect 'greening'; and data on accessibility of the land-use and water-body covariates are incorporated. A (non-Bayesian) hierarchical methodology is employed wherein elements of the total covariate portfolio are selected, arbitrarily, at different geo-spatial scales; and are, subsequently, regressed against the life-style indicator. Spatial auto-regression techniques are employed with ad hoc spatial weight and ad hoc covariate choice. Willingness-to-pay measures for access to greener areas are supplied and are used to make connection between 'determinants', 'lifestyle perception', and 'the value that individuals may be willing to assign to various correlates'. The present methodology could be used to extend the exercise. The covariate-selection and geo-spatial-weight aspects of the exercise and the (classical) hierarchical

model setting could be formalized in a natural, and perhaps more robust, systemic manner following well-advanced (Bayesian) protocols evolving from seminal ideas (in [40,95]).

In one of the most widely read articles to appear in *Land Economics*, [96] examines cost–benefit strategies in order to mitigate the impacts of loss habitat on endangered-species, extinction rates. They analyze potential biological-conservation, geographic-reserve sites, and select the 'best' sites enabling the highest density of terrestrial vertebrate conservation in Oregon, United States. The data include measures of species, their ranges, and the land values implicated by the proposed conservation areas. The authors of [96] locate cost-effective strategies—strategies that solve the problem of maximizing the total number of species protected for a given conservation budget—and establish profiles of the preservation-versus-cost-of-coverage by sequentially varying the proposed budgets. The exercise shares similarities with present endeavor. On the one hand, there is the notion of 'best'. On another, there is concern for the coefficients supplied to the mathematical programming exercise and the possibly that excluded correlates exist that may over-arch proceedings. Also, there is ultimate concern for the sensitivity of findings to another that takes full and formal account of these vagaries in predicting appropriate action (site choice) across the site-choice portfolio. In fact, the present methodology appears well suited and highly applicable to the type of analysis undertaken by [96], removing it from a programming aspect to one of fundamental empirical content in which the marginal likelihood plays a defining and fundamental role. The nexus of similarity lies in the stochastic incorporation of the ranking and selection protocol, which is well-suited to reanalysis. Also, it motivates one additional potential future enquiry.

Finally, we conclude this very brief and admittedly superficial tour of potentially related contributions with mention of [97,98]. The authors of [97,98] review 'state-of-the-art land-use change modelling tools', 'policy options', and 'research priorities'. They note that numerous models are available for enactment and that this plethora exemplifies diverse disciplinary background. Reviewing currently used models, they identify a total of *six concepts* they claim are important for land-use modelling, namely the level of (scale of) analysis applied; the presence of cross-scale dynamics; the particular driving forces; the existence of possible spatial interactions and neighborhood effects; the existence of temporal dynamics; and the overall level of integration. To this list we add two, namely formal accounting for the diverse probabilities affecting land-use policy decision-making, and formal accounting for pervasive model uncertainty in land-use pathway design.

The brief survey offered here detects indications of future directions, which time alone may confirm or refute as 'possible' or 'probable' or, better, 'fundamental' for informed land-use pathway design.

## 8. Conclusions

We commenced proceedings with some objectives, namely, demonstrating ranking-and-selection protocols for stochastic policy evaluation; showcasing routine extension of existing *MCMC* methodology; and incorporating due regard for inherent model uncertainty. One unexpected benefit arose, along with one unanticipated cost. The unexpected benefit is propagation of robust procedure for searching the covariate-threshold-geo-spatial model space; the unanticipated cost is refutation of previous work. The empirical results of our Ethiopian-highlands case study furnish strongest support for expanding local-breed, as opposed to cross-breed, herd size as the 'optimal' market-precipitating strategy evidencing that approximately eighty percent of the time this option is 'best'. Future research should assess the dependence of this conclusion on model-uncertain adaptations that the author, no doubt, has neglected.

The ready availability of a single-unifying quantity ([40]) bestows advantages but also confounds efforts by making accessible 'an embarrassment of riches' for model formulation and posterior probability checks. As in any investigation, there are a host of assumptions, which, with the benefit of hindsight, the investigator should wish to relax. This exercise is not immune to this criticism. Some possible fruitful directions are noteworthy.

First, there remains extension to fully incorporate, adequately, the full hierarchical structure of the survey. This structure is slightly non-conventional. Note that we have 68 households for which demographic data are collected; and that these measures are panel-invariant. However, the production data are recorded at each of the three visits during the one production year, yielding a 204-observation panel. At each of these visits the cow resources are recorded (note in particular that both ownership and the numbers milked are available). Also, at each of the three visits the sales data are recorded on that day, along with the production figure on that day, but that the sales figures are recalled from the preceding six days (production neither recorded nor recalled for those sales-recall days). Departing from the cross-sectional implementation of the 68 households and evaluating fully, the hierarchical structure beckons.

Second, in addition to the hierarchy, one should also like to allow for differing values of spatial correlation at each of the two survey sites—presently, the correlation measure is homogenized. This relaxation could greatly influence findings. Also, the author has casual empirical evidence that correlations may dramatically differ. So, heterogeneous spatial correlation beckons.

Third, current efforts are directed toward only one particular formulation of spatial correlation. However, nowadays, the literature is replete with alternative formulations. So, extensions to consider spatial-error, or, to consider spatial-Durbin model settings beckon.

Fourth, there is an innovation with respect to the Tobit censoring threshold that one should like to employ, which is to make the threshold dependent on covariates. Presently, the threshold is simply an unknown parameter. Thus, covariate-dependent thresholds formulations beckon, and with it, an additional covariate-selection exercise.

Fifth, (prompted by the constructive criticisms of two conscientious referees) our study, like others, makes one very strong, over-arching assumption that we have maintained throughout, but not tested. This assumption is that *a Bayesian model exists.* Matters pertaining to this philosophy are embedded in the exchangeability protocols espoused by [41,42,69,72,73], by [40,74], and others. Also, to the best of the author's knowledge, with one exception [99], these have yet to be enacted empirically. While these matters lie outside of present scope, they are obviously important and perhaps far-reaching. They are taken up, at one superficial level, employing the production data accompanying the present Tobit exercise in a companion paper. And that paper is available to the reader along with requests for this manuscript's on-line appendix.

Finally, drawing upon thematic methodological advance, while appealing, may be one step quite far removed from the realities of public policy decision-making throughout sub-Saharan Africa in general, and Ethiopia in particular. The reasons are manifold; they pertain to the vagaries of social-state governance; social disequilibria, including, in some settings, conflict; autocratic rule in certain regions; and the inherently disadvantaged standpoint of those human subjects likely to benefit most from the practicalities of the methodological advance. In Ethiopia, especially, the political economy of pro-poor livestock decision-making is observed ([100]) to be fettered by a number of features somewhat unique to the Ethiopian landscape [101–106]. In particular, livestock decision-making within policy realms is considered to be disaffected by the decades-long history of centralized governmental and administrative hierarchies; the incorrect view that livestock are mainly considered productive in their capacities as draught animals enabling mixed farming systems; inherently high transactions costs associated with the marketing and exporting of livestock and livestock products; the governmental view that unofficial cross border trade is sometimes viewed as 'contraband'; and the ongoing 'anti-pastoralist bias's towards pastoralism and pastoralism participants ([100]). More recently, the one over-arching public policy nexus likely to impact greatly farming systems in the near future concerns water management and energy production ([107]). Although more recent evidence paints a more positive outlook ([108]), it remains to be seen how gradual uptake of the new methodology instills pro-active public policy debate among prospective human-livestock policy commentators.

**Funding:** This research received no formal funding.

**Acknowledgments:** The data used in analysis were previously made available to the author during consultation at The International Livestock Research Institute, Addis Ababa, Ethiopia, 1998–2001. Philip Jones offered comments that led to significant improvements. The author is grateful to the International Livestock Research Institute and to Philip Jones but bears full responsibility for any error.

**Conflicts of Interest:** The author declares no conflict of interest.

## Appendix A

This manuscript contains MATLAB©-based computer protocols enabling readers to reproduce key results in the Markov-Chain Monte Carlo exercise; re-analyze data; and implement the search and ranking-and-selection procedures, with a minimum of effort. The computer code contains an over-arching executable file (jlgsiexec.m) which contains reference to eighteen sub-files, namely:

#1.  jlgsistart.m
#2.  jlgsiconfirm.m
#3.  jlgsipanel.m
#4.  jlgsicrossection.m
#5.  jlgsiglance.m
#6.  jlgsiselecttobit68.m
#7.  jlgsidesigntobit68.m
#8.  jlgsincmnlrm.m
#9.  jlgsincmnlrmcsa.m
#10. jlgsisearchtobit68.m
#11. jlgsifigureone.m
#12. jlgsifiguretwo.m
#13. jlgsifigurethree.m
#14. jlgsifigurefour.m
#15. jlgsifigurefive.m
#16. jlgsifiguresix.m
#17. jlgsifigureseven.m
#18. jlgsifigureeight.m

These eighteen files, executed sequentially, enable readying for analysis the Ethiopian household data; basic understanding of the marginal-likelihood criterion developed as part of the natural-conjugate, matrix-normal model; basic understanding of the covariate selection algorithm; basic imputation in the search algorithm; and reproduction of the key results, as developed and presented in the body of the manuscript. Specifically: #1. jlgsistart.m loads the file 'originaldata.mat' (supplied with the computer code) containing the Ethiopian household survey data; #2. jlgsiconfirm.m runs a confirmation check on the crossbreed and local-breed covariates; #3. jlgsipanel.m produces the full panel of two-hundred-and-four observations comprised of the three visits on the sixty-eight households; #4. jlgsicrossection.m forms the 'cross-section' by averaging across the panel; #5. jlgsiglance.m checks each column of the cross-section data for inconsistencies; #6. jlgsiselecttobit68.m selects the variables to be included in the Tobit censored regression analysis; #7. jlgsidesigntobit68.m designs the twenty-covariate and spatial weight matrices within which covariate, spatial-autoregressive and threshold selection occurs; #8. jlgsincmnlrm.m specifies, generating simulated data, the natural-conjugate matrix-normal linear-regression model, including the marginal-likelihood evaluation, as presented in [109] (p. 589, Equation A.29) and its basic-marginal-likelihood-identity decomposition as derived, initially, in [110] and exploited, subsequently, in [111]; [109] (p. 586, Equation A.5; and p. 587, Equation A.18); #9. jlgsincmnlrmcsa.m presents the fundamental covariate-selection algorithm based on the natural-conjugate matrix-normal regression presented in file #8, adapted from [109] and exploited subsequently; #10. jlgsisearchtobit68.m presents the search procedure generating subsequent graphics;

#11. jlgsifigureone.m produces text figure one; #12. jlgsifiguretwo.m produces text figure two; #13. jlgsifigurethree.m produces text figure three; #14. jlgsifigurefour.m produces text figure four; #15. jlgsifigurefive.m produces text figure five; #16. jlgsifiguresix.m produces text figure six; #17. jlgsifigureseven.m produces text figure seven; and #18. jlgsifigureeight.m produces text figure eight.

The time-consuming components of the executions are #10, #16 and #17, which absorb a few hours on a relatively modest hardware platform (an Intel® Core™2 Duo CPU E8499 @ 3.00GHz with 4.00 GB (3.84 GB usable) installed memory (RAM) running on a 64-bit Operating System).

Readers applying code are kindly requested to cite *This Issue* of the *Journal* as its source. Readers encountering problems executing code or wishing to establish extensions of this work for 'bespoke' analyses are encouraged to contact the author directly at the address for correspondence listed on the title-page.

Finally, although de-emphasized, the adaptation of the [89] algorithm used to formalize model search and, in particular, the results in Figure 5, is formalized on the notion of 'exchangeability' [40–42,69,72–74]. Informative background rading on exchangeability and its implications for statistical reasoning can be found in [112]. An example of the central idea in [112] in a related, empirical setting, using components of the Ethiopian household survey data; application to an 'assignment problem', beyond the scope of present analysis; and related digressions including the 'test' for a Bayesian model; are laid out in a related mimeograph completed during the process of the present investigation ([113]); the mimeograph is made available along with requests for this paper's bespoke computer code.

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
