# Peer review of "Sustainable Land-Use Pathway Ranking and Selection"

_sustainability, doi:10.3390/su12197881_

Round 1

Reviewer 1 Report

Thanks for the opportunity to do this review. The authors have carried out a high quality work. The manuscript is correctly developed, objectives have been set, the methodology is justified, the bibliography is good, etc. There are some aspects that nevertheless should be improved, so that the investigation will be polished. Authors are recommended:
- Justify in a more profound way the research gap on which they have focused, indicating if possible quotes from authors who recommend researching this problem.
- Further justify the social utility of this topic.
- In the conclusions section, the discussion about the results obtained is very scarce. This discussion cannot be limited to a repetition of the results, but must be contextualized and related to the underlying problem.
- Further input from the authors on the implications of their research is required, as well as public policy recommendations.

I think these changes will improve the work done.

Reviewer 2 Report

This study proposed the methodology for ranking and selecting sustainable ‘land-use pathways,’ arguing that the methodology is central to sustainable-land-use-policy prescriptions, providing essential innovation to assessments hitherto devoid of probabilistic foundation.

This study is interesting, however, there are some issues need to explain more clearly, such as following.
1. Please identify the weakness of the previous studies.
2. Please provide the related work.
3. The ranking and selecting problems are typical MCDM problems, why do you not employ the MCDM methods such as AHP or ANP?
4. Why do you employ the Markov-Chain, Monte-Carlo procedure ?
5. Please identify the contribution of this study.

Round 2

Reviewer 2 Report

This revised manuscript has addressed my concern.

I suggest to accept the paper.